# Liposome System for Encapsulation of *Spirulina platensis* Protein Hydrolysates: Controlled-Release in Simulated Gastrointestinal Conditions, Structural and Functional Properties

**DOI:** 10.3390/ma15238581

**Published:** 2022-12-01

**Authors:** Maryam Forutan, Maryam Hasani, Shirin Hasani, Nasrin Salehi, Farzaneh Sabbagh

**Affiliations:** 1Department of Food Science and Technology, Shahrood Branch, Islamic Azad University, Shahrood 3619943189, Iran; 2Department of Fisheries, Faculty of Fisheries and the Environment, Gorgan University of Agricultural Sciences and Natural Resources, Gorgan 4913815739, Iran; 3Department of Basic Sciences, Shahrood Branch, Islamic Azad University, Shahrood 3619943189, Iran; 4Department of Chemical Engineering, Chungbuk National University, Chungbuk 28644, Republic of Korea

**Keywords:** protein hydrolysate, liposomes, simulated gastric fluids, simulated intestine fluids, FTIR

## Abstract

This study aimed to evaluate the physicochemical, structural, antioxidant and antibacterial properties of chitosan-coated (0.5 and 1% CH) nanoliposomes containing hydrolyzed protein of *Spirulina platensis* and its stability in simulated gastric and intestine fluids. The chitosan coating of nanoliposomes containing *Spirulina platensis* hydrolyzed proteins increased their size and zeta potential. The fourier transform infrared spectroscopy (FT-IR) test showed an effective interaction between the hydrolyzed protein, the nanoliposome, and the chitosan coating. Increasing the concentration of hydrolyzed protein and the percentage of chitosan coating neutralized the decreasing effect of microencapsulation on the antioxidant activity of peptides. Chitosan coating (1%) resulted in improved stability of size, zeta potential, and poly dispersity index (PDI) of nanoliposomes, and lowered the release of the hydrolyzed *Spirulina platensis* protein from nanoliposomes. Increasing the percentage of chitosan coating neutralized the decrease in antibacterial properties of nanoliposomes containing hydrolyzed proteins. This study showed that 1% chitosan-coated nanoliposomes can protect *Spirulina platensis* hydrolyzed proteins and maintain their antioxidant and antibacterial activities.

## 1. Introduction

After warnings of the rapid population growth and food challenges in the middle of the twentieth century, many efforts have been made to introduce a cheap and easy alternative to protein, and algae has gained a high value [1,2]. *Spirulina platensis* is filamentous microalgae with a special importance due to its high amount of nutrients, especially protein [3,4] and thus can be a good option in this case.

Nowadays, many studies have been conducted to produce bioactive peptides from the hydrolyzed protein of marine organisms to produce a high value-added product [5,6]. In addition to functional and nutritional properties, these bioactive peptides have broad therapeutic effects [7,8,9]. Various methods are employed to separate peptides from proteins (hydrolysis), and enzymatic hydrolysis is known as an effective method in this field [10]. Notably, the enzyme on the functional properties of the peptide has a great effect on the size and hydrophilicity of hydrolyzates [11,12]. Bioactive peptides must remain active during digestion and absorption before entering the bloodstream to exert their physiological effects [13,14].

There are various methods to protect these peptides [15], and the use of microencapsulation technology can practically protect bioactive peptides against digestion. As lipid carriers, Liposomes are used to coat bioactive substances and food-drug. These colloidal vesicles are composed of polar lipids, especially phospholipids, which form bilayer spherical structures in the presence of water molecules [16]. Due to their polar lipids, they can encapsulate a wide range of compounds. Nano-liposomes provide more surface area compared to liposomes, due to their smaller particle size, higher solubility, improved bioavailability, controlled release, and more accurate delivery of materials to target areas. However, lipid-based delivery systems are not suitable in gastric acid, bile salts, and pancreatic lipase due to instability [16]. Mucoadhesive polymers such as chitosan can enhance the liposomal delivery of bioactive peptides. Thus, the stability of liposomal carriers and the efficiency of absorption in the gastrointestinal tract can be greatly increased by a layer of chitosan coating [17].

Previous studies have examined chitosan-coated nanoliposomes carrying bioactive compounds. They have shown that the coating of curcumin-carrying nanoliposomes with chitosan improved stability and decelerated the release of curcumin in the simulated gastrointestinal fluids [18,19]. Moreover, the chitosan-coated liposomes containing coenzyme Q10 and alpha-lipoic acid improved the antioxidant and antibacterial properties of these compounds [19]. Researchers found that chitosan-coated nanoliposomes, which carry the hydrolyzed protein of the orange seed showed the positive effect of encapsulation in liposomes in protecting, controlling, releasing, and maintaining the antioxidant activity of peptides [20]. However, there are a few studies that encapsulate the hydrolyzed protein of microalgae in liposomes and examine encapsulated hydrolyzed protein properties. Therefore, the present study investigated the antioxidant and antibacterial properties of uncoated and chitosan-coated nanoliposomes containing hydrolyzed protein of spirulina and analyzed its stability in simulated gastric and intestine media.

## 2. Materials and Methods

### 2.1. Materials

The microalgae *Spirulina platensis* was purchased from the noordaru company (Gonbad Kavous, Golestan, Iran). Alcalase enzyme (2.4 Anson units per gram) was purchased from Sigma-Aldrich, Inc. (St. Louis, MO, USA). Soy lecithin and glycerol were purchased from the Merck Company (Weiterstadt, Germany). The coating materials were the Chitosan with a medium molecular weight (75–85% degree of deacetylation) (Sigma-Aldrich). All other chemicals and reagents were of analytical grade. The reference strains of two bacteria including *Escherichia coli* (*E. coli*, CMCC 44825) and *Staphylococcus aureus* (*S. aureus*, ATCC 29213) were obtained from the Persian Type Culture Collection, Tehran, Iran. Distilled and deionized water was used for the preparation of all solutions.

### 2.2. Extraction of Spirulina Protein

Pure *Spirulina platensis* Powder was purchased and stored in a sealed plastic package at room temperature. First, 1 g of *Spirulina platensis* algae powder was added to 100 mL of distilled water and was shaken at 400 rpm for 30 min. Then, this mixture was sonicated (20 kHz, 25% power level, and 750 W) in an ice bath for 5 min (Sony Vibra-Cell, John Morris Auckland, Auckland, New Zealand). The sonicated mixture was then centrifuged (8000 rpm for 30 min) and the supernatant was collected. The supernatant-containing protein was desalted using ammonium sulfate (0–20% and 20–80%) salts and then dialyzed in ammonium bicarbonate buffer (100 mM) at pH: 8 for 8 h [21,22]. The second step of dialysis was carried out in distilled water for 2 h and then centrifuged at 5000 rpm for 15 min.

### 2.3. Enzymatic Hydrolysis

The extracted protein was redissolved in ammonium bicarbonate buffer at pH: 8 (2 mg/mL) and then the Alcalase enzyme was added (2%). This enzymatic reaction was performed at 45 °C for 10 h [23]. To stop the enzymatic activity, the reaction medium was heated at 90 °C for 10 min and then centrifuged at 7000× *g* for 20 min at 4 °C. After dissolving O-phthaldaldehyde in methanol (40 mg/mL) and beta-mercaptoethanol (100 μL), 10 μL of the sample or standard was mixed with 200 μL of this freshly prepared O-phthaldehyde solution and incubated in the dark for 2 min. The absorption was determined using spectrophotometry (Multipad Inspire Plate Reader, Perkin Elmer, Waltham, MA, USA) and the amino acid concentration resulting from the enzymatic hydrolysis was determined by drawing the L-leucine standard curve [24]. The degree of hydrolysis (DH) was calculated based on the concentration of extracted protein and the amino acid concentration.

### 2.4. Preparation of Nanoliposomes

The heating method [25] was employed to prepare raw nanoliposomes. Soy lecithin (5% *w*/*v*) and 14% hydrolyzed protein (10 mg/mL) were diluted in 2 mL of distilled water for 1 h at room temperature under atmospheric nitrogen pressure. This mixture was then increased to 10 mL by PBS containing glycerol (3%). The pH of this mixture was adjusted in the range of 7.3 using 2 M NaOH and was shaken at 60 °C and 1000 rpm and under atmospheric nitrogen for 60 min. After cooling the sample at room temperature, the raw liposomes were homogenized using a homogenizer at a pressure of 117.2 MPa to form a liposomal suspension [26].

### 2.5. Preparation of Chitosan-Coated Nanoliposomes

Chitosan solution was prepared by dissolving chitosan in 1% *v*/*v* in acetic acid and shaking it overnight at room temperature on a shaker and then filtering with a Whatman syringe filter (0.45 μm). This solution was added dropwise to liposomal suspension and shaken at room temperature (200 rpm for 1 h) and incubated overnight at 4 °C. Uncoated liposomes and coated liposomes with chitosan (0.5% and 1 *w*/*v*) was homogenized under high pressure. To get the smaller particle size of the liposomes, the suspension was subjected to a sonication process in an ice bath (7 min, 1 s on and 1 s off) using a probe (Sonicator, 200 UPS, Dr. Hieschler, Teltow, Germany).

### 2.6. Evaluation of Nanoliposome Properties

#### 2.6.1. Nanoparticle Size, Polydispersity Index (PDI), and Zeta Potential

The mean size and polydispersity index of liposomes were determined with the dynamic light scattering (DLS) method and photon correlation spectroscopy (PCS). The measurements were performed at a temperature of 25 °C using a zeta sizer (Malvern, MA, USA) equipped with a helium-neon laser with a scattering angle of 173°. In addition, the zeta potential of liposomes was measured with the same device at 25 °C.

#### 2.6.2. Encapsulation Efficiency

Firstly, 0.5 mL of liposome solution was combined with 1 mL of acetone and centrifuged at 3 °C and 5000× *g* for 30 min. The supernatant-containing unencapsulated peptides was separated and placed in an oven to evaporate the solvent at 60 °C. Then, the remaining dry matter was combined with 5 mL of distilled water and its protein content (unencapsulated peptides) was determined with the Lowry method. On the other hand, to determine the concentration of all peptides in liposomal suspension, 0.5 mL of primary liposome solution was mixed with 1 mL of triton ×100 (0.06 g/100 g) to completely dissolve phosphatidylcholine and destroy the wall. Therefore, the amount of encapsulated peptides was obtained from the difference between the total peptides present in the liposomal suspension and the unencapsulated peptides. Finally, encapsulation efficiency was obtained by calculating the ratio between the amount of encapsulated peptides and the total amount of peptides present in the liposomal suspension [27].

#### 2.6.3. Fourier Transform Infrared Spectroscopy (FTIR)

To evaluate the chemical structure, the nanoliposomes were pulverized by freeze-drying. Then, each sample was mixed with potassium bromide in a ratio of 1 to 100 and formed into a disk using a pressure machine. Finally, FTIR spectroscopy of coated and non-coated liposomes was performed using the FTIR spectrophotometer with a scan range of 400–4000 cm^−1^ [20].

#### 2.6.4. Evaluation of Antioxidant Activity

To measure the DPPH inhibitory activity of nanoliposomes, coated and non-coated liposomes containing 14% hydrolyzed protein with a concentration of 0.2–2 mg/mL were placed in a water bath at a temperature of 100 °C for 5 min. Then, the obtained sample (1.5 mL) was combined with 1.5 mL of DPPH solution (freshly prepared in ethanol). The mixture was then vortexed and kept in the dark for 30 min and centrifuged at 4000× *g* for 10 min. The absorbance (A) was measured at 517 nm with a visible light spectrophotometer. Inhibitory activity was calculated using the following formula [28]:DDPH inhibitory activity% = (A_control_ − A_sample_/A_control_) × 100(1)

To measure the ABTS radical inhibitory activity of the samples, ABTS solution (7 mM) was first combined with 2.5 mM potassium sulfate. The resulting mixture was placed in the dark at room temperature for 12 to 16 h. ABTS solution was diluted with PBS (0.2 M) to achieve absorption of 0.7 at 734 nm. Nanoliposomes were heated at 100 °C for 5 min to release the encapsulated peptides. Then, 20 μL of different concentrations of the samples were mixed with 980 μL of ABTS reagent and stored for 10 min at 30 °C in a dark place. The absorbance of the samples was measured at 734 nm. The inhibitory of radicals was calculated using the following formula [29]:ABTS inhibitory activity% = (A_control_ − A_sample_/A_control_) × 100 (2)

#### 2.6.5. Morphology of Nanoliposomes

The morphology of nanoliposomes was evaluated using transmission electron microscopy (TEM) and the negative staining method. The suspension containing nanoliposomes was diluted with distilled water, combined with ammonium molybdate solution (2% *w*/*v*), and placed at room temperature for 3 min. A drop of this solution was poured onto a formvar/carbon-coated grid and placed at room temperature for 5 min. Finally, after drying the grade, nanoliposomes were imaged using a transmission electron microscope [30].

#### 2.6.6. The Stability of Nanoliposomes in Simulated Gastrointestinal Fluids

To evaluate the stability of liposomes, simulated gastric fluid (SGF) with pH: 1.2, simulated intestinal fluid (SIF) with pH 6.8, and phosphate buffer saline (PBS) with pH: 7.4 were prepared. Then, 200 μL of liposomal suspension was diluted with 10 mL of these fluids and incubated at 37 °C for 2 h in SGF, 4 h in SIF, and 6 h in PBS [31]. Nanoparticle size, Zeta potential, and PDI were evaluated according to the methods mentioned above.

#### 2.6.7. The Release of Encapsulated Protein in Simulated Gastrointestinal Fluids

In this experiment, the pepsin and pancreatin were removed from the SGF and SIF, respectively. Nanoliposomal suspension (20 μL) was diluted with 1 mL of the fluid and incubated at 37 °C and 80 rpm for 2 h for SGF, 4 h for SIF, and 24 h for PBS in separate microcentrifuge tubes. These tubes were centrifuged at 45,000 rpm [31] and the amount of protein in the supernatant was measured according to the method mentioned above.

#### 2.6.8. Evaluation of Antibacterial Activity

*Escherichia coli* (*E. coli*, CMCC 44825) and *Staphylococcus aureus* (*S. aureus*, ATCC 29213) were purchased and prepared. The broth dilution method was adopted to evaluate the minimum inhibitory concentration (MIC). The bacteria were diluted with Müller-Hinton Broth medium at 5 × 10^5^ CFU/mL. The liposomes were dissolved in DMSO (dimethyl sulfoxide) to make a solution of 100 mg/mL and then diluted (0.5, 1, 2, 4, 8, 16, 32, 64, 128, and 256 mg/mL) by the culture medium. These dilutions (100 μL) were added to 96-well plates containing 100 μL of the desired bacteria and incubated at 37 °C for 24 h. Then, the amount of light absorption at 600 nm was measured with a spectrophotometer and the rate of inhibition of bacterial growth was calculated according to the following formula [32]:MIC_50_% = (A_control_ − A_sample_/A_control_) × 100 (3)

To determine the minimum bacterial concentration (MBC), Müller Hinton agar solid culture medium was used. Diluted pre-prepared bacteria were incubated with liposomes to prepare the liposome concentration of 4, 8, 12, 16, 20, 24, 28, and 32 mg/mL. The tubes were incubated at 37 °C for 2, 6, and 24 h. Serial dilutions were prepared in saline solution (0.9%) and 1 mL of each dilution was spread evenly on the solid culture medium. After 18 h of incubation at 37 °C, the CFUs per dilution were counted. MBC was calculated as the minimum nanoliposome concentration that resulted in a 99.9% reduction in bacterial growth compared to the control [32].

### 2.7. Statistical Analysis

One-way analysis of variance (ANOVA) was performed using SPSS (ver.20) software. Analytical data were obtained from analyses of three samples for each treatment in chemical assays. Differences among mean values were examined with Duncan’s test (*p* ≤ 0.05) significance level.

## 3. Results

### 3.1. The Concentration of Extracted Protein and DH

The amount of extracted protein was 108 mg/L. After 10 h of exposure of this protein to Alcalase, the mean percentage of enzyme hydrolysis was 14%.

### 3.2. Nanoparticle Size, PDI, Zeta Potential, and Encapsulation Efficiency and Morphology of Nanoliposomes

Non-coated nanoliposomes were the smallest in size and significantly different from the other treatments (Table 1). In terms of PDI, nanoliposomes coated with 1% chitosan did not show a significant difference with uncoated and coated nanoliposomes with 0.5% chitosan. The zeta potential of coated nanoliposomes was not significantly different from each other. On the other hand, coating nanoliposomes with 1% chitosan resulted in a significant reduction in encapsulation efficiency (*p* < 0.05).

The results of transmission electron microscopy (TEM) evaluation of non-coated and coated nanoliposomes are shown in Figure 1. According to the figure, the highest size is related to nanoliposomes with 1% chitosan coating and the lowest size is related to non-coated nanoliposomes. In addition, the nanoliposomes had a spherical shape and a smooth surface.

### 3.3. FTIR Spectroscopy

According to Figure 2A, the most important absorption peaks in spirulina hydrolyzed protein include those at wavelengths of 3346 cm^−1^ (OH), 3240 cm^−1^ (OH), and 1715 cm^−1^ (C=O), and 1636 cm^−1^ (COOH). Figure 2B shows the absorption peaks of the non-coated nanoliposome carrying 14% hydrolyzed protein, in which the peak at 3239 cm^−1^ corresponds to the absorption peak of 14% hydrolyzed protein at 3240 cm^−1^. On the other hand, this peak appeared in the FTIR spectrum for nanoliposomes coated with 0.5% chitosan at 3249 cm^−1^ (Figure 2C). Figure 2D shows the absorption peaks of nanoliposomes coated with 1% chitosan. Among the absorption peaks observed, the peaks of 3245 cm^−1^ and 1705 cm^−1^ corresponded to the absorption peaks of 3240 cm^−1^ and 1715 cm^−1^, respectively, for 14% hydrolyzed protein.

### 3.4. Antioxidant Properties

Low concentrations of hydrolyzed proteins (0.4 and 0.8 mg/mL) had a significant difference in DPPH radical inhibition compared with coated and non-coated nanoliposomes. However, as the concentration increased, this difference narrowed so that no significant difference was observed in DPPH radical inhibition between the hydrolyzed protein and 1% chitosan-coated nanoliposome at higher concentrations of hydrolyzed protein (Figure 3a). The observed trend in DDPH radical inhibition was similarly observed in ABTS radical inhibition (Figure 3b). However, the difference between the hydrolyzed protein and nanoliposomes in ABTS radical inhibition was more obvious and no significant difference was observed between the three nanoliposomes in ABTS radical inhibition at low concentrations of hydrolyzed protein. The results also showed that the difference between groups in the ABTS radical inhibition decreased with higher protein concentration and there was no significant difference between coated nanoliposomes in protein concentrations of 1.6 and 2 mg/mL.

### 3.5. The Stability of Nanoliposomes in Simulated Gastrointestinal Fluids

According to Figure 4a, nanoliposomes coated with 1% chitosan had the highest size stability and non-coated nanoliposomes had the lowest size stability in the simulated media. In other words, a significant difference in the size of non-coated nanoliposomes compared to the control (initial size of nanoliposomes) in the simulated media indicates a reduction in size and instability. However, these significant differences are also observed in nanoliposomes coated with 0.5% chitosan. On the other hand, the simulated intestinal fluid had the greatest effect on reducing the size of nanoliposomes.

As shown in Figure 4b, changes in the PDI of coated and noncoated nanoliposomes in the simulated fluid have a similar trend and differ only significantly from their control (initial PDI) (*p* < 0.05).

Changes in the zeta potential of coated and non-coated nanoliposomes in simulated gastric and intestinal fluids are shown in Figure 4c. The trend of changes in the zeta potential of nanoliposomes in the simulated media is almost the same among the treatments. However, 1% of chitosan-coated nanoliposomes appear to be more stable than others.

### 3.6. The Protein Release Profile from Nanoliposomes

As shown in Figure 5a, the highest rate of peptide release from nano-liposomes in the simulated intestinal fluid was observed in non-coated nanoliposomes and the lowest rate in 1% chitosan-coated nanoliposomes. On the other hand, the release rate of nano-liposomes increases with the increasing time of exposure to the intestinal fluid (*p* < 0.05).

Figure 5b shows the results of the release profile of nano-liposomes in the simulated gastric fluid. The process observed for the release of nanoliposome peptides in this medium was similar to the one observed in the intestinal fluid. However, the percentage of peptide release in the simulated gastric medium after exposure time (120 min) is less than that in the simulated intestinal medium.

Figure 5c shows the results of nanoliposome peptide release in PBS during 5 to 24 h. Although the process of releasing peptides from nano-liposomes is similar to the simulated gastric and intestines, the time interval is longer than the previous two fluids. After 24 h of exposure to PBS, the peptide release from non-coated nanoliposomes reaches up to 47%, while it reaches 37% in 1% chitosan-coated nanoliposomes.

### 3.7. Antibacterial Properties

The results for the MIC of 14% hydrolyzed protein and coated and non-coated nanoliposomes are shown in Table 2. The MIC of spirulina hydrolyzed protein for *Escherichia coli* is lower than that of *Staphylococcus aureus*. In addition, the encapsulation of hydrolyzed protein leads to an increase in the MIC and a decrease in its antibacterial properties. However, 1% chitosan-coated nanoliposomes did not differ significantly from the hydrolyzed protein in the MIC for the two bacterial species (*p* > 0.05). On the other hand, there is no significant difference in the MIC between the two bacterial species in non-coated nanoliposomes.

In contrast, the MBC of hydrolyzed protein for *Escherichia coli* did not have a significant difference with *Staphylococcus aureus* (*p* > 0.05). In addition, the encapsulation of hydrolyzed protein leads to more MBC and less antibacterial properties. However, coating the nanoliposomes with 1% chitosan largely compensated for this reduction. On the other hand, the antibacterial properties of non-coated nanoliposomes were significantly different between the two bacterial species, and their antibacterial properties increased compared to *Staphylococcus aureus* (*p* < 0.05).

## 4. Discussion

### 4.1. Nanoliposome Size, PDI, Zeta Potential, and Encapsulation Efficiency and Morphology of Nanoliposomes

The size of the nanoliposome was enlarged by coating it with chitosan and increasing the chitosan concentration. Enlarging nanoliposomes by raising the chitosan concentration confirms the formation of the chitosan layer on the surface of liposomes [33]. Microscopic images of non-coated and coated nanoliposomes and their morphological examination showed larger nanoliposome size after coating and elevated chitosan concentration, which further confirms the formation of a chitosan layer on the nanoliposome. Coating the nanoliposome with chitosan results in the formation of a hydrogen bond between the polysaccharide and the phospholipid [34]. The nanoliposome size is a major indicator of bioavailability, and solubility of bioactive compounds within the nanocarrier system. In previous studies, the average size of non-coated nanoliposomes containing hydrolyzed fish proteins was 263 nm [28], and the size of chitosan-coated nanoliposomes containing olive leaf extract was less than 100 nm [35]. In the present study, nano-liposomes had a spherical shape and a flat surface, as found in earlier studies [36]. Some studies also showed that nano-liposomes can have non-spherical shapes through the application of high shear force such as high-pressure microencapsulation [37]. Non-spherical nanoparticles have fewer lipid layers compared to spherical surfaces. Many properties of lipid nanoparticles such as physical and chemical stability, microencapsulation efficiency, loading capacity, the position of bioactive compounds inside the nanoparticle, and release rate of the compound significantly affect the shape of nanoparticles [28].

In the present study, the PDI of the coated and non-coated nanoliposomes ranged from 0.2 to 0.3, indicating a uniform dispersion of nanoliposomes. When this index has a high value (≈1), it indicates that the sample has large and accumulated particles. Li et al. reported that increasing particle size leads to greater PDI, and the chitosan coating causes a reduction in the PDI [33]. In the present study, it seems that the effect of increasing the size of nanoliposomes on the PDI was greater than the effect of the chitosan coating so the PDI in 0.5% chitosan-coated nanoliposomes showed a significant rise. The zeta potential indicates reactions between the liposome and the chitosan and is another confirmation of the coating of the liposome with the chitosan. In other words, the zeta potential indicates the stability of suspended particles in a suspension based on the electrostatic repulsion force between them. When the zeta potential of a suspension mixture containing nanoparticles is less than −30 mV and more than +30 mV, the suspension is highly stable due to the electrostatic repulsion force [38]. In the present study, the zeta potential range for non-coated nanoliposomes was in the range of 9.4 ± 0.75, and in 1% chitosan-coated nanoliposomes it was in the range of 14 ± 1.4, which indicates potential instability of nanoliposomes. However, chitosan coating reduced this instability to some extent. Nevertheless, the zeta potential of chitosan-coated nanoliposomes in the present study is acceptable compared to other similar studies, including the zeta potential of nanoliposomes containing hydrolyzed fish muscle protein, reported as 5.5 mV [27]. Additionally, the zeta potential of nanoliposomes shifted to positive values due to coating as well as increasing the chitosan concentration. The zeta potential of non-coated nanoliposomes depends on the phospholipid composition and it consequently depends on the concentration of chitosan in chitosan-coated nanoliposomes [38]. In the present study, the coating of nanoliposomes with chitosan enhanced the zeta potential, which could be due to the association of more cationic polymers on the surface of liposomes [23]. On the other hand, no significant change in zeta potential was observed with a higher chitosan percentage, which may be due to the larger nanoliposome size [39]. The leveling off of zeta potential is supposed to be where the anionic liposomal surface is saturated with the cationic polymer coating. Following the addition of the chitosan, the particle size of nanoliposomes increased and the zeta potential changed from negative to positive values, demonstrating that cationic chitosan successfully coated the liposomes. These results are in agreement with those of Mohammadi et al. [40], who found that covering of liposomes containing protein hydrolysate from *Spirulina plantensis* with chitosan cationic polymer by electrostatic interaction increased the particle size of liposomes and the coated liposomes represent a positive charge.

Chitosan coating increases the zeta potential resulting in electrostatic inter-particle repulsion, reducing the system’s unstable mechanisms, such as aggregation, sticking, coagulating, and ultimately improving the physical stability of the conduction system.

Most studies with polysaccharides or protein carriers did not publish the zeta potential of the encapsulated protein hydrolysates and peptides. This knowledge is especially useful in assessing the effects of the processing techniques used for these carriers, such as freeze drying, on the encapsulated bioactive compound stability.

The maximum encapsulation efficiency in the current study was about 85%, which is an acceptable percentage in comparison with 73% to 92% in the study of Sarabandi et al. [28] and 71% in the study of Lee et al. [41]. The encapsulation efficiency of 1% chitosan-coated nanoliposomes showed a significant decrease compared to the other two groups, which was probably due to the aggregation of 1% chitosan-coated liposomes resulting from the establishment of nanoliposome–chitosan–nanoliposome bridges. As stated in the case of particle size and PDI, this finding can be related to the instability of the membrane structure of nanovesicles and their aggregation and agglomeration. These results are in accordance with those reported by Mohammadi et al. [42].

In similar studies, maximum EE of the salmon (about 71%) and rainbow trout (about 80%) peptides were obtained in the chitosan-coated nanoliposomes [26,43]. The results obtained in the present study are probably due to the protective effect of the coating layers on peptides adhering to the liposome surface. On the other hand, increasing the concentration of chitosan to more than 1% chitosan leads to a lower encapsulation, probably due to a similar charge of chitosan to the peptide, which can be replaced in the liposome [43].

### 4.2. FTIR Spectroscopy

An FTIR test was run to investigate the interaction of 14% hydrolyzed protein with nanoliposomes, and the results showed effective interactions. This test is a suitable method for evaluating the structure of the acyl group in phospholipids. In addition, the changes in the structure of fats and physicochemical processes can be found by analyzing the changes in frequency and wavelength [28]. This test can assess the potential reaction between phosphatidylcholine and hydrolyzed proteins in the nano-liposomes. In the present study, a wavelength shift from 3240 (OH) to 3239 was observed in the non-coated nanoliposome carrying 14% hydrolyzed protein, indicating both the presence of the hydrolyzed protein in the non-coated nanoliposome and its reaction with the lipid structure. Previous studies have shown that the absorption peaks in the initial range of the spectrum are caused by sharp fluctuations in the hydroxyl (OH) and amine (NH) groups and possibly the formation of a hydrogen bond between these groups in peptide and lecithin [27].

In non-coated nanoliposomes carrying hydrolyzed protein, the flexibility of acyl chain in lipid membrane led to CH2 symmetric stretching at 2824 cm^−1^, which is observed in empty nanoliposomes at 2852 cm^−1^ [44]. In addition, the peak at wavelengths of 1740 cm^−1^ in non-coated nanoliposomes carrying hydrolyzed protein indicates the C=C bond stretching between the hydrocarbon chain and the polar group, result from the peak of 1754 cm^−1^ in empty nanoliposomes [18]. Moreover, the absorption peak of 1066 cm^−1^ results from the absorption peak of 1049 cm^−1^ in empty nanoliposomes and indicates the P-O stretching. In chitosan-coated nanoliposomes, the peak of 2824 cm^−1^ underwent an extensive shift and appeared at 2924 cm^−1^, indicating the reaction of chitosan with the lipid structure. The peak of 1754 cm^−1^ in the empty nanoliposome shifted to 1738 cm^−1^ in the chitosan-coated nanoliposome, indicating interactions through hydrogen bond formation between the carbonyl group (C=O) of phosphatidylcholine and the active compound. These results lead to the hypothesis that the hydrolyzed proteins are probably located in a polar region [27], which confirms our hypothesis that the zeta potential of nano-liposomes is positive.

### 4.3. Antioxidant Properties

The results showed that the antioxidant activity of proteins slightly decreased after encapsulation into the liposomes. However, in the samples coated with 0.5% chitosan, more than 45% of the DPPH radical scavenging activity was maintained. The results of the present study showed that by increasing the percentage of chitosan, the antioxidant properties of liposomes improved, which could be due to the unique antioxidant properties of chitosan and the synergistic effect with protein [43]. It can be found that protein encapsulation in the liposome forms a hard layer around the protein and limits the protein’s availability with free radicals, thereby showing less antioxidant effect. The use of the two methods (DPPH and ABTS assays) in the present study was due to the diversity of oxidation processes and the antioxidant activity of hydrolyzed proteins, which invalidates the employment of a single method for evaluating antioxidant properties [45]. The relationship between the concentration of hydrolyzed protein and the percentage of DPPH free radical scavenging has also been reported in a previous study, in which the DPPH free radical scavenging increased with higher hydrolyzed protein concentration [46]. Amino acids glycine and leucine play a major role in the antioxidant properties of peptides. The mechanism of this action by the amino acid glycine is related to its side chain, which is a single hydrogen atom and makes the peptide highly flexible for access and antioxidant properties [47].

ABTS is a relatively stable radical and is easily inhibited by antioxidants. Furthermore, its inhibition test is a suitable method for measuring the antioxidant activity of reducing compounds and aqueous phase radicals. Ramezanzadeh et al. showed that encapsulation of peptides derived from fish skin resulted in a significant reduction in ABTS radical inhibition, which was moderated by a higher concentration of chitosan coating. This trend was also recorded for DPPH radicals [43], confirming the results of the present study. The effect of encapsulation of protein hydrolysate via the enzymatic hydrolysis of the flaxseed protein [28], rainbow trout skin [43], and sea bream scales collagen [27] on the antioxidant activity of nanoliposomes was determined in other studies. In this research, preserving the antioxidant activity of peptides after encapsulation into the nanoliposomes was reported.

### 4.4. The Protein Release Profile from Nanoliposomes

Coating nanoliposomes with 1% chitosan leads to greater size stability, zeta potential, and PDI compared to non-coated nanoliposomes. Positively charged chitosan coatings form an ionic bond with negatively charged liposomes, leading to mucoid adhesion and enhancing their stability. Hasan et al. reported that polysaccharides (chitosan) can inhibit lipid digestion and lead to the difference in the release rate of curcumin from chitosan-coated nanoliposomes and non-coated nanoliposomes [18]. Coumo et al. reported that the three simulated fluids have some differences (e.g., bovine serum albumin, enzymes, pH, and thus the positive or negative charge of their compounds), leading to different behaviors of nanoliposomes in the release of curcumin [39]. These results can be extended to the current study in terms of release rate and effect on zeta potential, nanoliposome size, and PDI. However, explaining the exact mechanism requires further studies and further experiments in future research.

### 4.5. Antibacterial Properties

The present study showed that the encapsulation of the hydrolyzed protein led to an increase in the MIC and the MBC. Sun et al. isolated the antibacterial peptide SP-1 from spirulina with a MIC of 8 mg/mL for *Escherichia coli* and 16 mg/mL for *Staphylococcus aureus* [46]. The antibacterial activity of peptides is highly dependent on their helical alpha structure, size, hydrophobicity, and hydrophilicity. The lower antibacterial activity of the encapsulated hydrolyzed protein in the present study may be due to the inaccessibility of hydrolyzed protein and its encapsulation. On the other hand, chitosan coating improved the antibacterial properties of nanoliposomes containing hydrolyzed proteins, which can also be due to the antibacterial properties of chitosan.

The increase in antibacterial properties of non-coated nanoliposomes against *Staphylococcus aureus* is an interesting result that could be related to its ability to produce perforating toxins. This bacterium secretes proteins that act as cytotoxins, the most predominant of which are the toxins that cause holes in cell membranes (membrane-perforating toxins) [47]. Cui et al. investigated the antibacterial activity of clove oil encapsulated in nanoliposomes against *Escherichia coli* and *Staphylococcus aureus* [48]. The results of their study showed that the antibacterial property of this oil in the form of nanoliposomes was greater against *Staphylococcus aureus*, which was attributed to the production of membrane-perforating toxin. In the present study, the bacterium released the hydrolyzed spirulina protein from the nanoliposome, thereby enhancing its antibacterial properties against *Staphylococcus aureus*. It may not hold for coated nanoliposomes, and the chitosan coating protects the nanoliposomes against this toxin. However, further studies are needed to confirm the findings of this study in this respect.

## 5. Conclusions

The results of the present study showed that microalga *Spirulina platensis* is considered a source of protein in terms of having bioactive peptides with unique properties. The results showed that liposomal nanocarrier can be observed as one of the effective systems for the nanoencapsulation of bioactive compounds such as protein hydrolyzate. High encapsulation efficiency, particle size distribution, and zeta potential of nanoliposomes presented high stability of chitosan-coated nanoliposome. The profile of protein release showed that nanoliposomes, especially chitosan-coated nanoliposomes, controlled the release of proteins successfully. Furthermore, the results showed that nanoliposomes can protect the trapped bioactive compounds and improve the stability of the protein in an SGF and SIF medium. The results clearly showed that pure protein hydrolyzate and coated nanoliposomes have a strong antioxidant activity by inhibitory activity against free radicals. In addition, similar antimicrobial activity was observed between pure protein hydrolyzate and nanocapsules coated with 1% chitosan. Overall, this study considers that the use of nanoliposome-containing protein derived from microalgae can be a natural compound with preservative properties used in functional food.

## Figures and Tables

**Figure 1 materials-15-08581-f001:**
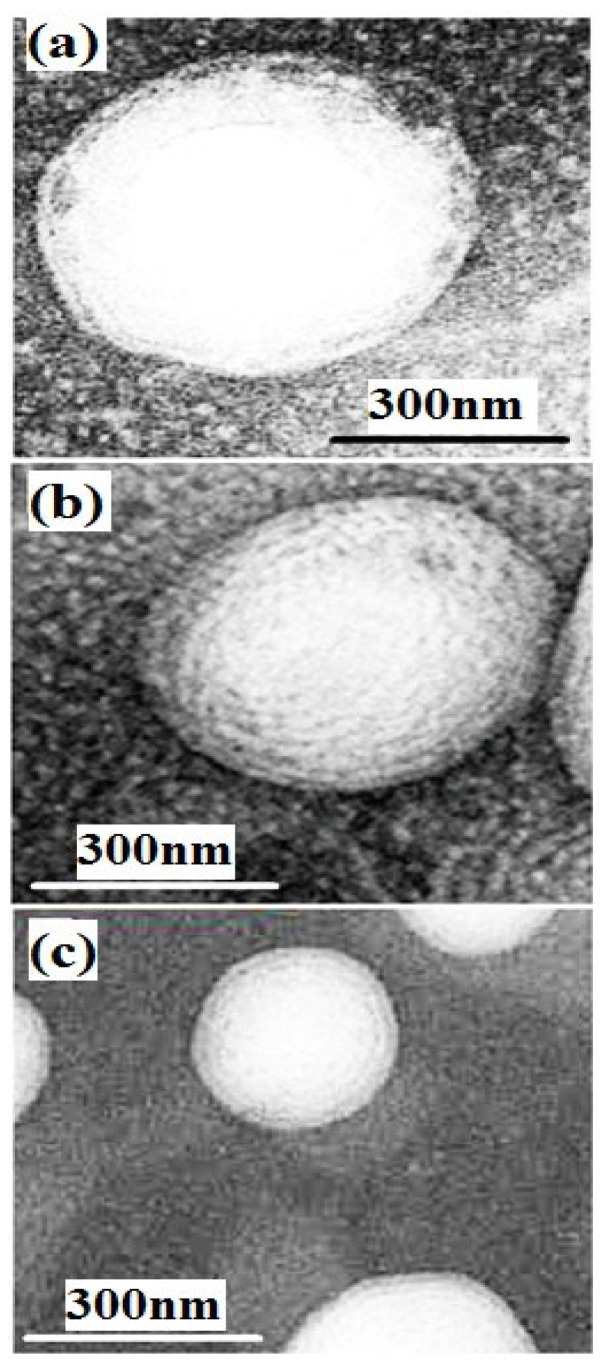
Morphology of nanoliposomes, (**a**) nanoliposomes coated with 1% chitosan, (**b**) nanoliposomes coated with 0.5% chitosan, (**c**) non-coated nanoliposomes.

**Figure 2 materials-15-08581-f002:**
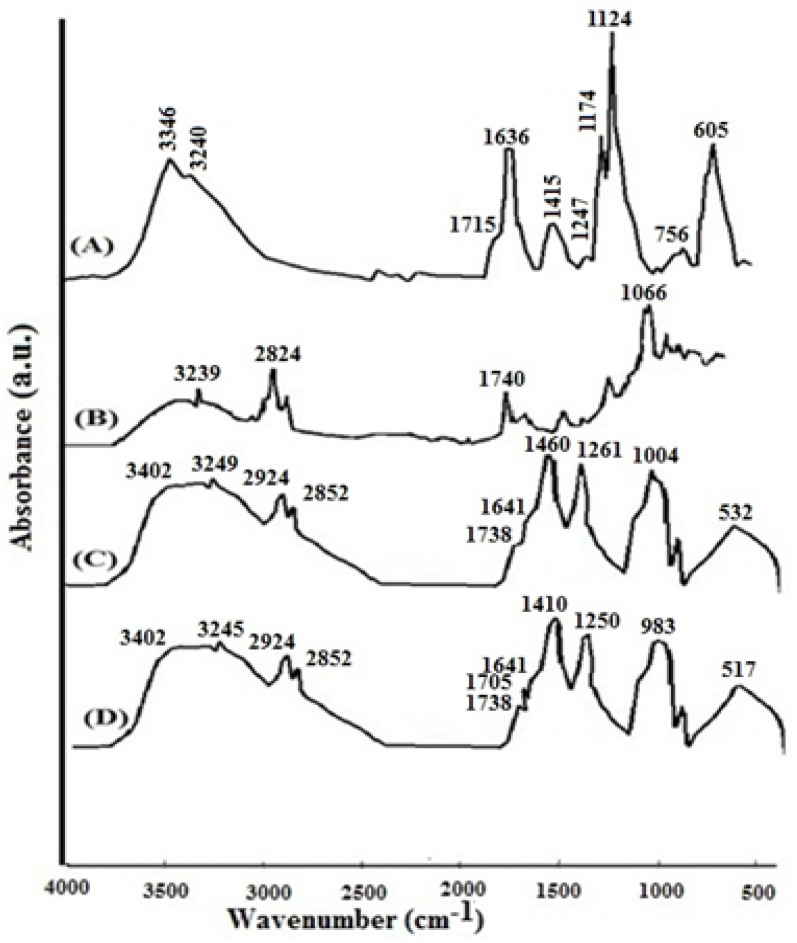
FTIR spectroscopy, (**A**) FTIR spectrum for 14% hydrolyzed protein, (**B**) FTIR spectrum for non-coated nanoliposomes containing 14% hydrolyzed protein, (**C**) FTIR spectrum for 0.5% chitosan-coated nanoliposomes containing 14% hydrolyzed protein, (**D**) FTIR spectrum for 1% chitosan-coated nanoliposomes containing 14% hydrolyzed protein.

**Figure 3 materials-15-08581-f003:**
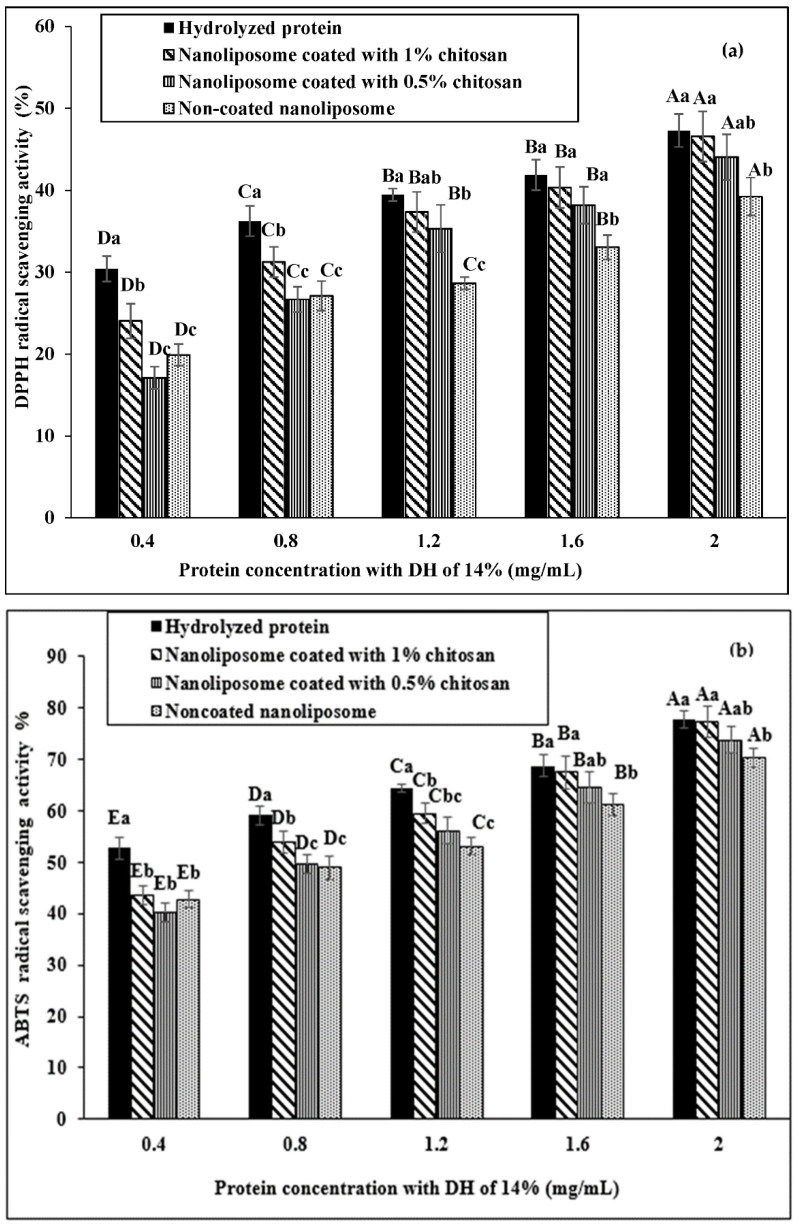
DPPH (**a**) and ABTS (**b**) radical scavenging capacity by hydrolyzed protein, non-coated nanoliposomes, 0.5% chitosan-coated nanoliposomes, and 1% chitosan-coated nanoliposomes. Different lowercase letters indicate a significant difference between groups at one concentration and different uppercase letters indicate a significant effect of the concentrations on the DPPH and ABTS scavenging.

**Figure 4 materials-15-08581-f004:**
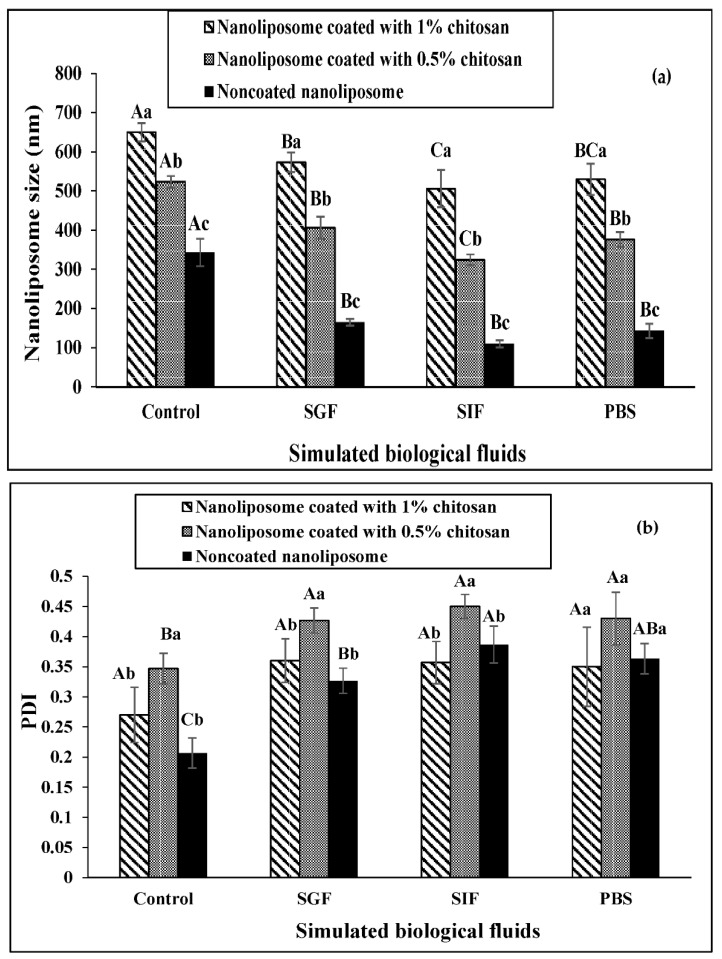
Size stability (**a**), PDI (**b**), and zeta potential (**c**) of nanoliposomes in simulated gastric fluid (SGF), intestine fluid (SIF), and PBS. Identical uppercase letters indicate no significant difference between one nanoliposome in different fluids compared to the control (initial measure) and identical lowercase letters indicate no significant difference between different nanoliposomes in one fluid.

**Figure 5 materials-15-08581-f005:**
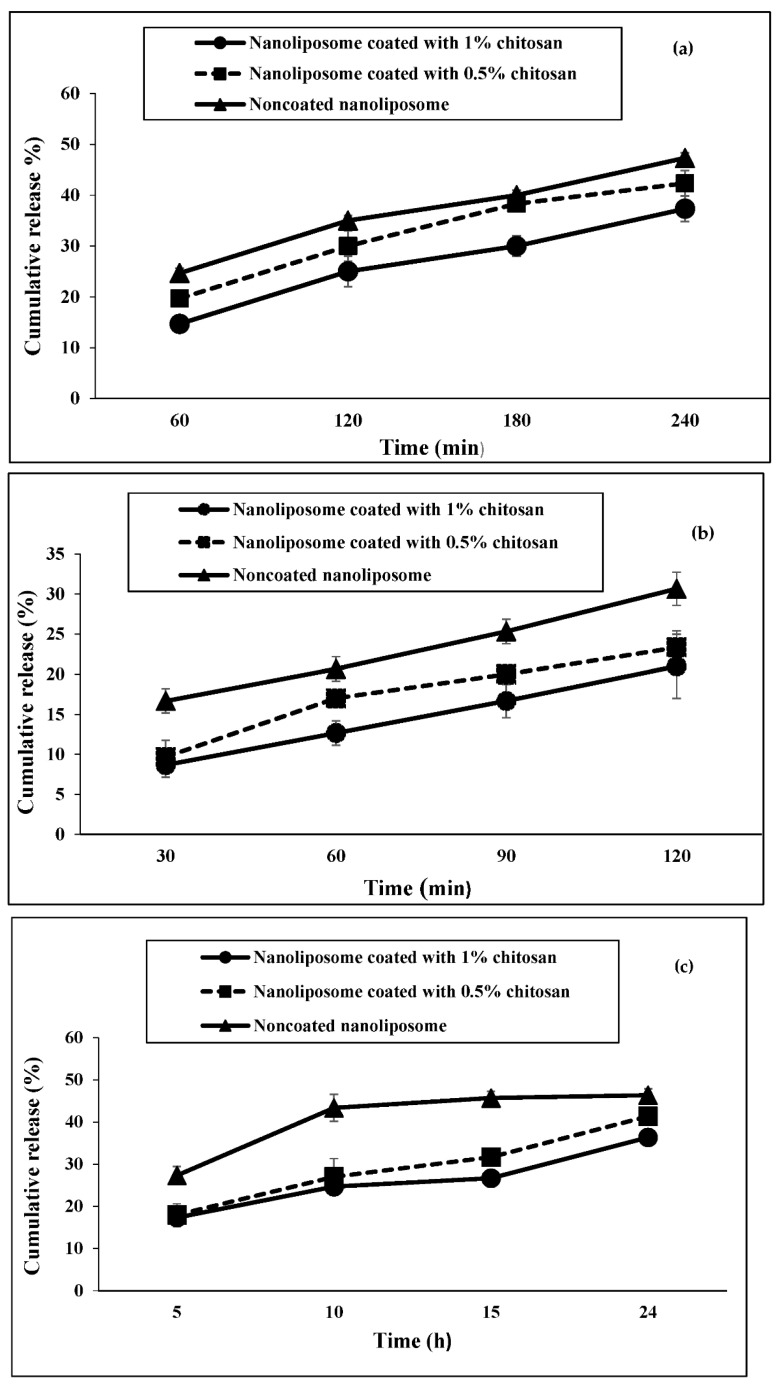
Release of nanoliposome peptides in different simulated media over different periods, (**a**) intestine, (**b**) gastric, (**c**) PBS.

**Table 1 materials-15-08581-t001:** Particle size, zeta potential, PDI, and microencapsulation efficiency of nanoliposomes containing 14% hydrolyzed protein.

	Chitosan Coated Nanoliposome (1%)	Chitosan Coated Nanoliposome (0.5%)	Non-Coated Nanoliposome
Nanoliposome size (nm)	644 ± 39.50 ^a^	552 ± 35.38 ^b^	348.33 ± 35.47 ^c^
Zeta potential (mV)	14.00 ± 1.41 ^a^	13.26 ± 0.66 ^a^	9.43 ± 0.75 ^b^
PDI	0.27 ± 0.04 ^ab^	0.30 ± 0.03 ^a^	0.20 ± 0.02 ^b^
Encapsulation efficiency (%)	73.10 ± 2.70 ^b^	84.70 ± 2.60 ^a^	83.80 ± 2.55 ^a^

Identical lowercase letters indicate no significant differences between non-coated nanoliposomes, nanoliposomes coated with 0.5% chitosan, and nanoliposomes coated with 1% chitosan (*p* > 0.05).

**Table 2 materials-15-08581-t002:** MIC and MBC of 14% hydrolyzed protein and coated and non-coated nanoliposomes.

Treatment	MBC (mg/Ml)		MIC (mg/mL)	
	*E. coli*	*S. aureus*	*E. coli*	*S. aureus*
Hydrolyzed protein	12.0 ± 4.0 ^Ab^	10.6 ± 1.3 ^Ab^	3.3 ± 1.1 ^Bc^	6.6 ± 2.3 ^Ab^
Non-coated nanoliposome	22.6 ± 2.3 ^Aa^	17.3 ± 2.3 ^Ba^	16.0 ± 0.0 ^Aa^	16.0 ± 0.0 ^Aa^
Nanoliposome coated with 0.5% chitosan	14.6 ± 2.2 ^Ab^	12.0 ± 0.0 ^Ab^	6.6 ± 1.3 ^Bb^	8.0 ± 0.0 ^Ab^
Nanoliposome coated with 1% chitosan	9.3 ± 2.1 ^Ab^	8.0 ± 0.0 ^Ab^	3.3 ± 1.1 ^Bc^	6.6 ± 2.3 ^Ab^

Identical lowercase letters indicate no significant difference between groups for each bacterium (MIC and MBC, separately), and identical uppercase letters indicate no significant differences between the two bacterial species in each group.

## Data Availability

The data presented in this study are available on reasonable request from the corresponding author.

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
