# Peer review of "Liposome System for Encapsulation of Spirulina platensis Protein Hydrolysates: Controlled-Release in Simulated Gastrointestinal Conditions, Structural and Functional Properties"

_materials, 2022, doi:10.3390/ma15238581_

Round 1
Reviewer 1 Report
The work proposed by the authors seems to me to be well focused on solving problems that we will encounter in the future, but some changes, modifications and clarifications have to be made.
According to the liposome preparation method, all of them are formed with hydrolyzed proteins, that is, all the synthesized nanocarriers have the drug (hydrolyzed protein) with different encapsulation efficiency. The authors perform size, PDI and zeta potential studies of these, but must compare the uncharged nonosystems, only the nanoliposome. And then if they carry out the study they have done with the loaded systems.
The authors could check the zeta potential of liposomes without chitosan but loaded with 14% protein, it is surprising that this is positive.
The results shown in figures 4, how much time elapses between the preparation of the samples and the measurements.
With regard to the release profiles, the authors must take into account that the release takes place at all three pHs independently. since if they were combined, the amount of drug released would be 105%, when the amount encapsulated is only 80%.
Table 1 and 2 have many errors, first the units of measure for the zeta potential are missing. check the way of expressing the values in table 1 and 2, since they are not being applied well. since the errors are expressed with only one significant figure unless this is a 1 that is expressed with two significant figures, and the position occupied by the last figure of the error is the last significant figure of the measurement. for example: 0.27 ±0.04 is well expressed, 9.43±0.75 is not, it would be 9.4±0.8. 0.2 ±0.02 is also wrongly expressed, it would be 0.20 ±0.02.
Authors should write title text behind the graph for all graphs that explains what is represented.
Author Response
Dear Editors-in-Chief
We would like to thank the reviewer for a careful and thorough reading of this manuscript and for the thoughtful comments, which help to improve the quality of this manuscript. Our response follows

Reviewer 2 Report
The submitted manuscript “Liposome system for encapsulation of Spirulina platensis protein hydrolysates: controlled-release in simulated gastrointestinal conditions, structural and functional properties" by Froutan et al. presents the physicochemical, structural, antioxidant and antibacterial properties of chitosan-coated nanoliposomes containing hydrolyzed protein of spirulina and its stability in simulated gastric and intestine fluids. The Authors have chosen an appropriate set of methods to characterize the obtained liposomes. High encapsulation efficiency, particle size distribution, and zeta potential of nanoliposomes presented high stability of the nanoliposome. This manuscript should be of interest and utility to workers in a number of branches of the field.
However, some changes and explanations are required:
1) Paragraph 2.6.2 - Why was the liposome solution combined with acetone? What is the role of acetone here?
2) Paragraph 2.6.6 - How exactly were the simulated gastric fluid and simulated intestinal fluid prepared?
3) Paragraph 2.6.8 - Why were the liposomes dissolved in DMSO? Doesn't DMSO kill bacteria by itself?
4) Table 1 - The number of significant places in the standard deviation and the value it describes should be the same.
5) Table 1 – What is “)nm(“? Similar brackets appear in Table 2 and Figures.
6) The diameters of the liposomes shown in Figure 1 should reflect the data given in Table 1 and should be close to the average values given in Table 1. For example, Table 1 shows that the smallest size of the non-coated liposomes was 310 nm (308.86 nm to be precise), whereas the liposome shown in Fig. 1c has a diameter significantly smaller than 300 nm. Similar doubts arise from the comparison of the diameters of the other liposomes shown in the figure and the table. To sum up, the illustrative photos need to be matched to the data in the table.
7) page 6 - There is no reference to Fig. 3a in the manuscript.
8) Paragraph 6.8 - I cannot agree with the statement: “the release rate of nanoliposomes”.
9) Paragraph 4.1 - I cannot agree with the statement: “The nanoliposome size is a major indicator of colloidal stability”. This would be correct if it referred to zeta potential, not size.
10) Paragraph 4.1 - The maximum PDI value is 1. It is generally accepted that PDI values between 0.3 and 1 are indicative of system instability and inhomogeneity. So assuming that a value of 0.34 indicates a uniform dispersion of nanoliposomes is questionable
11) Paragraph 4.1” “These results lead to the hypothesis that the hydrolyzed proteins are probably located in a polar region [27], which confirms our hypothesis that the zeta potential of nanoliposomes is positive.” This conclusion regarding the positive zeta potential charge is unclear to me. Please explain this in more detail.
12) The editorial side needs attention - there is sometimes unjustified use of capital letters in the middle of a sentence or starting a sentence with a lowercase letter.
I recommend the presented manuscript for publication only if all the accusations will be included in the new version.
Author Response

(The authors gave the same response as above.)

Reviewer 3 Report
Dear Editor
The manuscript entitled "Liposome system for encapsulation of Spirulina platensis protein hydrolysates: Controlled-Release in simulated gastrointestinal conditions, structural and functional properties". The manuscript is interesting, but some considerations should be needed attention.
Abstract
Please replace "Spirulina" with "Spirulina Platensis" and write it in Italic in the whole manuscript.
Please don't use the abbreviation for the first time in the manuscript e.x. " The FT-IR " and "DPI".
Introduction
"Hasan et al. showed that", "Zhao et al. found that", and "Mazloomi et al. found that" I think it doesn't really matter who reports it. So, instead of expressing in this way, I suggest that you highlight what is described and what is intended to be given to the reader.
The authors mentioned, "However, there is no study to encapsulate the hydrolyzed protein of microalgae in liposomes and examine encapsulated hydrolyzed protein properties". Please revise this sentence through the following manuscripts https://doi.org/10.3390%2Fantiox10121953 https://doi.org/10.1016/j.foodchem.2022.133973
Also, the manuscript could be substantially improved by relying on these manuscripts.
Materials and methods
Replace the subtitle "2.2. Extraction of Spirulina protein" with "2.2. Extraction of Spirulina Platensis protein". Also replace "Pure Spirulina Powder (Spirulina platensis)" with "Pure Spirulina Platensis powder" and "First, 1 g of Spirulina algae" with "First, 1 g of Spirulina Platensis algae".
Please add a reference for the preparation of Chitosan-coated nanoliposomes.
Results
Why the results of "3.1. The concentration of extracted protein and DH" is not present in table or figure?
Conclusion
Please remove "according to the results", "the results showed that", and "the results clearly showed that" and rewrite the section of the conclusion.
Figure captions
Please add all details e.x. in figure 1: Evaluation of the nanoliposomes morphology using transmission electron microscopy (TEM) and the negative staining method.
Figure 1: Please replace "(B) nanoliposomes" with "(b) nanoliposomes" as presented in the figure.
References
Please add the doi.
Author Response

(The authors gave the same response as above.)

Round 2
Reviewer 1 Report
Thank you very much for the answers, I only have two comments:
Review the zeta potential value of the liposome alone, because with a value of 0 it would practically not be soluble in aqueous solutions. On the other hand, the authors propose that the union between chitosan and liposomes occurs mainly by electrostatic interactions "that can be attributed to the union of chitosan with a positive charge to vesicles with a negative charge (Hasan et al., 2016)". but if we analyze the zeta potential values of the liposome alone or loaded with the peptide, these are neutral or positive. How is the interaction with chitosan justified?
The authors have only modified the values that I commented were wrong, and I only put a couple of examples for the application of the error rule, I am going to say how many values are wrongly expressed and to expose the error rule again. in table 1 there are 9 wrongly expressed values and in table 2 there are 6. "the errors are expressed with only one significant figure unless this is a 1 that is expressed with two significant figures and the position occupied by the last figure of the error is the last significant figure of the measurement."
Author Response
Thank you for the comments. The responses are attached.

Reviewer 2 Report
It's ok now.
Author Response
Thank you for considering the manuscript carefully.